# Effects of Outdoor Access and Indoor Stocking Density on Behaviour and Stress in Broilers in the Subhumid Tropics

**DOI:** 10.3390/ani9121016

**Published:** 2019-11-22

**Authors:** Rubi Sanchez-Casanova, Luis Sarmiento-Franco, Jose Segura-Correa, Clive J. C. Phillips

**Affiliations:** 1Facultad de Medicina Veterinaria y Zootecnia, Universidad Autónoma de Yucatán, Km 15.5 Carretera Mérida-Xmatkuil, Apdo. 4-116, Itzimna, Mérida, Yucatán 97100, Mexico; rcsan84@gmail.com (R.S.-C.); jose.segura@correo.uady.mx (J.S.-C.); 2Centre for Animal Welfare and Ethics, University of Queensland, White House Building (8134), Gatton Campus, Gatton, QLD 4343, Australia; c.phillips@uq.edu.au

**Keywords:** outdoor access, stocking density, broilers, behaviour, tropical regions, heterophil-lymphocyte ratio, corticosterone, stress, welfare

## Abstract

**Simple Summary:**

The stressors affecting chickens in farming systems differ from those confronting their wild ancestors; thus, the ability to cope with an outdoor environment may have been modified. There is a growing body of literature that recognises the importance of allowing chickens to express their natural behaviour by providing outdoor access. However, studies investigating the effects of offering outdoor access to housed meat chickens in the tropics are scarce. Therefore, the effects of allowing chickens outdoor access, with two indoor space allowances, on their growth, behaviour and responses to stress were assessed. Some blood parameters indicated that outdoor access reduced stress, but the behavioural effects of providing outdoor access depended on indoor stocking density. Birds at the high stocking density with outdoor access were found standing for longer periods of time and those at the high stocking density and no outdoor access were observed walking less and lying more than the other three groups. It is concluded that there were welfare benefits of outdoor access, principally in terms of increased activity, but slower growth resulted in lighter birds at the end of the study.

**Abstract:**

Studies investigating the welfare of commercial-line broiler chickens raised in houses with outdoor access in the tropics are scarce, and none have investigated whether responses vary according to indoor conditions. Hence, we assessed the effects of providing outdoor access at two indoor stocking densities on broiler chickens’ growth, behaviour, stress responses and immunity in a tropical region of Mexico. One hundred and sixty chickens were assigned to one of four treatments in a factorial design: with or without outdoor access and low or high stocking density indoors. Ad libitum sampling was used to build a purpose-designed ethogram. Scan sampling was used to record the number of birds engaged in each activity of this ethogram, both indoors and outdoors. Heterophil/lymphocyte (H/L) ratio and serum corticosterone levels were tested in weeks four and six of age. When the birds were 42 days old, they were slaughtered, and the bursa and spleen harvested and weighed. In an interaction between stocking density and outdoor access, birds at the high stocking density with no outdoor pens spent the least time walking and preening and more time lying (*p* < 0.05). Birds given outdoor access foraged more, but only at indoor low stocking densities (*p* < 0.05). Outdoor access reduced heterophil/lymphocyte ratio, indicating reduced stressor response. Birds with low stocking density indoors and outdoor access appeared more responsive to stressors, with elevated corticosterone and reduced spleen and bursa weights (*p* < 0.05). There were welfare benefits of outdoor access, principally in terms of increased activity, which were reflected in slower growth in the birds with outdoor access.

## 1. Introduction

There is growing demand for poultry meat worldwide, particularly in developing countries partly in response to population growth and growing affluence [1]. In response, poultry industries have intensified, adopting high stocking densities and rapid growth genetics to maximise output and profitability. Crowded living space can have a significant positive impact on farmer incomes, which generally increase with stocking density [2]. However, decrease in space allowance is likely to have detrimental effects on the welfare of the birds [3,4,5]. Given that the microclimate for chickens in most tropical conditions accords with their evolutionary environment, outdoor access could potentially improve their welfare without greatly increasing costs but it is still necessary to develop standards that ensure the outdoor area is properly functioning as a form of enrichment for the birds [6]. In particular, the possibility of predation, sporadic feed availability and transmission of diseases from wildlife must be recognised, but on the positive side, it could provide a valuable source of nutrients for poultry, especially since tropical ecosystems have a vast natural potential for production of forage plants.

Chickens behavioural responses to outdoor access systems stem from their jungle fowl ancestry [7]. These responses can be considered a result of decision-making processes that take into account the nature and the strength of the internal physiological state and external environmental signals that determine the probability of each behaviour being performed [8,9]. Nevertheless, the stressors affecting chickens in the human-controlled systems differ from those that confronted their wild ancestors, thus the ability to cope with different stressors may have been modified [10]. Quantifying the amount of time displaying certain behaviours gives some indication of their importance [11], providing evidence of the animals’ welfare [12]. Physiological indicators, such as the heterophil to lymphocyte (H/L) ratio and corticosterone concentration in blood serum, can also provide welfare information. Heterophils tend to increase and lymphocytes decrease in response to stress, but this avian immune response takes some time, measured in hours to days, to initiate, while a corticosterone response usually occurs within minutes [13,14,15,16]. Physical indicators, such as the weight of lymphoid organs, are related to stressor response through immune activation and suppression [17]. 

There is a growing body of literature that recognises the importance of allowing chickens to express their natural behaviour by providing outdoor access [18,19,20,21,22,23,24]. However, reports of the effects on the behaviour and welfare of commercial-line broiler chickens raised in housing conditions with outdoor access in the tropics are scarce. Therefore, the objective of this study was to assess the effects of housing system and stocking density on broilers’ behavioural repertoire, stress indicators and performance.

## 2. Materials and Methods

The study was conducted at the poultry farm of the Campus de Ciencias Biológicas y Agropecuarias (CCBA-UADY), located in the central region of Yucatán, Mexico. At an altitude of 10 m.a.m.s.l., the climate is typical of the warm subhumid tropics, with rain mostly in summer, a total annual rainfall of 999 mm, and a mean annual temperature of 28 °C, potentially increasing to 40 °C in spring and summer [25]. The winter season (December–February) was selected for this study to avoid, as far as possible, adverse welfare impacts of temperatures above 30 °C and relative humidity levels above 80% for the housed groups of birds without outdoor access [26].

### 2.1. Study Design

The experiment used a two-factor (outdoor access and stocking density) factorial design. One hundred and sixty male chickens were randomly assigned to one of four treatments: low stocking density with outdoor access (LO); high stocking density with outdoor access (HO); low stocking density indoors (without outdoor access) (LI); high stocking density indoors (without outdoor access) (HI). Each treatment had four replicates (pens). Of the 40 birds in each treatment, these were allocated at random to four groups (replicates) of ten birds. Two different enclosure sizes were used to achieve the two stocking densities. This constant group size has been showed to have only small effects on behaviour in terms of observed nearest-neighbour distances and amount of space used, compared with larger group sizes [27]. In stocking density experiments, either the group size must be kept constant and the total area varies, or the group size differs, and the total area can be kept constant. We chose the former as group size variation at these low levels was deemed more likely to induce changes in behaviour than varied total area. However, it has been demonstrated that total area available can have more effects on locomotion and use of space than group size, but only with a tripling of total area, not the doubling in our study [27]. Hence it is acknowledged that stocking density effects may be confounded with total area for the group in this study. 

### 2.2. Animals, Husbandry and Facilities

The day-old Hubbard Flex male chicks for this experiment were purchased from a commercial hatchery. All birds were initially placed inside a circular reception area with cardboard walls of 0.50 m height and wood shavings as bedding material. Bell-type drinkers and initial tray feeders were provided. Bell-type drinkers were chosen over others since they provide a higher water availability and promote natural drinking behaviour [28]. Lighting program was 24L:0D for the first seven days. At seven days of age, ten birds were randomly allocated to each of 16 naturally ventilated pens. After an adaptation period of seven days (i.e., when the birds were 14 days old), outdoor access was allowed for birds in the LO and HO treatments. Lighting program from day 15 to 21 was 14L:10D and 16L:8D from day 22 to the end of the experiment. The facilities were designed to allow the access to an average of 11 h of natural light a day (6:00–17:00 h). The remaining light hours were provided by using one 40 W incandescent bulb (15 lumens/watt) per building.

Feed and water were provided ad libitum. A three-phase feeding program was used, consisting of a starter diet from day 1–14 of age, grower diet from day 15–28, and finisher diet from day 29–42. The ingredients and nutrient composition of the diets met recommendations for Hubbard Flex chickens [29]. The body weight of all chickens was recorded weekly as the average per treatment. Individual birds were not identified for this purpose. On day 42, birds were weighed for the last time and the experiment ended.

The poultry facility was established in two portal-framed buildings each of dimensions 6 × 6 m. The roof was of galvanised steel sheets. Pens were arranged inside to minimise position effects in both horizontal directions, although the treatments with outdoor access were necessarily on one side of the building. The stocking density of the birds at the end of the experiment was approximately 30 kg/m^2^ for the high stocking density treatments, as recommended for commercial lines in hot weather conditions [29], and half that for birds in the low stocking density. Maintaining the same number of birds per pen (10), the low stocking density had 5 birds/m^2^, with a treatment enclosure size of 2 m^2^, and the high stocking density 10 birds/m^2^, with a treatment enclosure size of 1 m^2^. Pens were constructed of galvanised chicken wire mesh to a height of 1.20 m. Four had dimensions of 1.50 × 0.67 m (high stocking density) and four had dimensions of 1.50 × 1.34 m (low stocking density), as seen in Figure 1. Extra space (0.23 m^2^/pen) was provided to compensate for the area occupied by the feeder and drinker. For the outdoor access treatments LO and HO, each pen had an outdoor area of dimensions 1.5 × 5 m, which the birds could enter through a pop-hole of dimensions 0.50 high × 1.50 m wide, via a ramp of 0.50 × 1.50 m. Fences were of galvanised chicken wire mesh to a height of 1.2 m. The eight outdoor areas (5 × 1.50 m^2^ for each pen) were covered with natural vegetation (mostly *Pennisetum ciliare* and *Leucaena leucocephala*), as seen in Figure 2. 

### 2.3. Behaviour Recording

Behaviour observations started when the birds were 21 days old and concluded on day 41, prior to slaughter at 42 days old. All 160 animals were observed both in the pens and outdoor areas. Prior to the main experiment, an ad libitum sampling was conducted in similar conditions during weeks 3–6 of two production cycles to determine an appropriate ethogram and recording procedure. During these preparatory observations, behaviour was continuously recorded from 06:00 to 17:00 h, one day a week [30], by direct observation and by using a digital camera (Nikon^®^ COOLPIX L810, Nikon Corporation, Tokyo, Japan). Notes and 88 h of digital video recordings were combined to construct an ethogram of ten mutually exclusive behaviours, as seen in Table 1). 

During the experiment, birds in both the conventional and outdoor access groups were observed by the same researcher. Individual birds were not identified for this purpose but the numbers of birds engaged in each behaviour were manually recorded during the scan samples, performed at 10 min intervals three times a day (7:00–8:00; 12:00–13:00 and 16:00–17:00 h) for one day/week when birds were 3, 4 and 5 weeks old. For outdoor access treatments, behaviour was recorded without considering if birds were inside or outside the pen. It was not possible to complete the intended observation in week 6, since some animals exceed 3 kg of body weight and started to show mild signs of leg disorders. Video recording was used to crosscheck the behaviour at 10 min intervals and increase the accuracy of the counts [19,31,32]. The sole observer was blind to all aspects of treatments and intraobserver reliability, expressed as Pearson correlation coefficient, was 0.99 [30]. Indoor and outdoor temperatures were measured with a portable weather station (AcuRite^®^ Weather Environment System, model 01057RM, Chaney Instrument Co., Lake Geneva, WI, USA) throughout the study. For outdoor access treatments, indoor temperatures were measured at different points of the pens, far from the pop-holes.

### 2.4. Stress Indicators

The day after the behavioural observations on weeks 4 and 6, eight birds per treatment (2 birds/pen) were taken to a separate room to collect 3 mL of blood from the brachial vein. Sample collection was completed within one minute postcapture to avoid handling effects on physiological parameters [33]. Each sample was placed in two individually identified tubes, with and without anticoagulant for H/L ratio and serum corticosterone concentration determination, respectively. 

Whole blood was smeared on slides, which were dried, fixed and stained [34]. Smears were examined in oil immersion under a light microscope at 1000× magnification. The count of the various cell types was made on a total of 100 leukocytes. Absolute and relative blood cell counts were obtained. Absolute heterophil and lymphocyte counts were used to calculate H/L ratios [35].

Sera for corticosterone analysis were obtained by centrifugation for 15 min at 1000× *g*. Corticosterone concentrates were measured by enzyme-linked immunosorbent assay (ELISA) using a commercial kit (Corticosterone ELISA Kit ADI-900-097, Enzo Life Sciences Inc., Farmingdale, NY, USA). All samples were run in duplicate. Absorbance was measured at 450 nm on an ELISA microplate reader (BioTek^®^, model ELx800, BioTek Instruments, Inc., Winooski, VT, USA), then a correction was made by manually subtracting the mean optical density of the blank wells from all readings, since the ELISA kit product manual recommended reading the optical density at 405 nm. Interassay and intra-assay coefficients of variation were 29.6% and 13.5%, respectively. The smallest detectable concentration was 17.6 pg/mL.

Chickens were individually weighed before slaughter. The spleen and bursa of Fabricius of eight animals randomly selected per treatment were removed and weighed. The relative weight (RW) of the organs to body weight was determined as a ratio [17].

### 2.5. Statistical Analyses 

The pen was the experimental unit for all analyses. A split-split-plot design was used to test the effects of treatments and week of age on behaviour, H/L ratio, serum corticosterone levels, bird weight and relative weight of lymphoid organs. The main plot corresponded to housing system (outdoor access or indoors) and stocking densities (low: 5 animals/m^2^ or high: 10 animals/m^2^) combinations, with repeated measures (weeks 3, 4 and 5 for behaviour, 4 and 6 for H/L ratio and serum corticosterone) as subplots. Behaviour was analysed as the percentage of time performing a given activity in each pen, with seven observations at three times of the day for each recording day. As no significant differences were detected between the three times of the day, data were pooled, providing one value per behaviour per pen. Values for dustbathing were mathematically manipulated by taking square roots of the values, so that the residuals most closely approximated a normal distribution. Mock fighting and grooming were rarely observed, and it was not possible to transform to achieve normality or heteroscedasticity of residual plots. Descriptive data only is reported for mock fighting. For grooming, a Fisher’s exact test was used to measure differences between systems and densities. H/L ratio and serum corticosterone levels also breached the assumption of normal distribution of residuals, so a log_10_ transformation was applied. Data were analysed by fitting a mixed-effects model of Minitab 17 (Minitab Inc., State College, PA, USA, 2014), with pen as the random term and week as the repeated measure. Performance records were analysed considering week of age separately. The assumption of ANOVA that residuals are normally distributed was tested using the Anderson–Darling test. Least square means were calculated by using the EMMEANS subcommand of SPSS Statistics 20 (IBM Corp., Chicago, IL, USA, 2011). A Tukey post-hoc test was used to discriminate significant differences between pairs of means when significant differences were detected overall. A *p*-value of ≤0.05 was considered significant in all analyses.

### 2.6. Ethical Standards

The experiment described in this report complied with all relevant legislation in Mexico and was approved by the Bioethics Committee of the Campus de Ciencias Biológicas y Agropecuarias-Universidad Autónoma de Yucatán, CB-CCBA-D-2018-001.

## 3. Results

### 3.1. Behaviour

Average temperatures (°C) and relative humidities (%) recorded during the behavioural observations are presented in Table 2. 

The effects of system, stocking density and week of age on the behaviour of the broilers are shown in Table 3, with interactions between system and density in Table 4.

#### 3.1.1. System and Stocking Density Effects

Significant interactions between system and stocking density were observed, as presented in Table 4. Locomotion and preening were decreased in birds at the high stocking density in indoor pens, and there was a tendency for feeding to be also reduced in this treatment (*p* = 0.06). If there was an outdoor area, the high indoor density increased standing, compared with the low density.

#### 3.1.2. Bird Age Effects

A significant interaction between system and week of age was evident for foraging, as seen in Table 3. Birds foraged more frequently on week 5 than 3 or 4, if outdoor access was provided, compared with no outdoor access, as seen in Figure 3. The interactions between density and week of age were significant for lying and foraging: chickens in the high density treatments spent less time lying (66.04 ± 6.44 vs. 69.35 ± 3.52, *p* = 0.01) in week 5 than week 4, and foraged less (1.72 ± 0.94 vs. 2.26 ± 0.94, *p* = 0.006) in week 4, as seen in Figure 4. Significant effects of interaction between density, system and age were only evident for drinking. Chickens of the HO treatment drank more often in week 5 of age (9.01 ± 0.75) and less in week 4 (2.98 ± 1.20) (*p* = 0.04).

Grooming was observed more frequently in the HO treatment (n = 8, 3.81% of chickens performing this behaviour during 210 min of observations) than the HI treatment (n = 2, 0.95%) in week 4 (*p* = 0.007). Mock fighting was observed only in the LO (2.38% of 210 min of observations), LO (0.95%) and LI (1.90%) treatments in weeks 3, 4 and 5, respectively.

### 3.2. Stress Indicators

Serum corticosterone levels were increased in week 6 compared with week 4 of age, as seen in Table 5, but only when outdoor access was provided (1748.8 ± 1508.8 vs. 891.1 ± 630.1; *p* = 0.01), as seen in Figure 5. It was also increased for birds at low density on week 6 (1786.6 ± 1428.9), compared to week 4 (1163.2 ± 1048.1) (*p* = 0.02), as seen in Figure 6.

A significant interaction demonstrated that the H/L ratio was particularly reduced by outdoor access in birds at low density on week 6 (outdoor access 1.44 ± 0.94, no outdoor access 3.21 ± 0.95, *p* = 0.01), as seen in Table 6. Heterophil number was also higher in week 6 than week 4, when no outdoor access was provided (67.1 ± 6.16 vs. 43.4 ± 9.85, *p* < 0.001) and at high density (61.6 ± 6.43 vs. 42.3 ± 8.15, *p* = 0.004), as seen in Table 6. Lymphocyte number increased with outdoor access in birds kept at low density, during week 6 (40.4 ± 11.39), in contrast to chickens kept indoors (22.8 ± 4.83) (*p* = 0.001), as seen in Table 6.

Spleen weight and spleen relative weight were greater in chickens raised indoors, as seen in Table 7. Bursa weight was higher in birds housed indoors, with relative weight being particularly higher in the LI treatment (0.14 ± 0.03), compared to LO treatment (0.11 ± 0.04) (*p* = 0.046).

### 3.3. Body Weight

Chickens raised indoors were heavier on week 3 (1.09 ± 0.06 vs. 1.05 ± 0.06; *p* = < 0.001), 5 (2.70 ± 0.18 vs. 2.60 ± 0.19; *p* = 0.002) and 6 (3.39 ± 0.21 vs. 3.28 ± 0.23; *p* = 0.001), compared with those with outdoor access, and those kept at low density were heavier on weeks 4 (1.90 ± 0.13 vs. 1.84 ± 0.11; *p* = 0.004) and 5 (2.71 ± 0.18 vs. 2.60 ± 0.17; *p* = < 0.001), compared to those housed at high density, as seen in Figure 7 and Figure 8. No significant interaction effects were found between system and density.

## 4. Discussion

### 4.1. Effects on Behaviour

Since outdoor access systems have not been commonly implemented in tropical areas of Latin America, the development of a catalogue of behaviours to assess the welfare of chickens was an initial objective of this project.

Locomotion was much reduced in the HI treatment, with there being no difference between the other three treatments. Locomotion plays an important role in leg health, which is facilitated by both walking and running [36]. Physical activity increases both the thickness and density of the cortical bone and the diameter of the tibiotarsus diaphysis. The extent depends on the level of activity [32]. This deleterious effect of stocking density and age on locomotion has been observed previously [37,38]. Dawkins et al. [9] concluded that, although high stocking densities affect the welfare of birds, density itself is, within limits, less important for welfare than other factors of the environment in which broilers are housed. However, Buijs et al. [39] found that chickens are willing to work for larger floor space allowances. Therefore, different pen sizes could also have exerted an effect on locomotion, since it has been found that greater enclosures sizes promote higher rates of movement and activity [40]. Restrictions on both activity and expression of natural behaviour occur when floor space is insufficient [3,4,5,41,42,43]. Additionally, pen size is confounded with stocking density in this experiment, with larger pen sizes potentially increasing bird locomotion and interindividual distance, as noted in the Materials and Methods section [27,40,44].

The greatest number of chickens were found lying in the HI treatment. Although lying is good for welfare if birds are comfortable, with healthy broilers spending up to 76% of their time lying [11], several studies have indicated that a reduction in space decreases movement, as well as being associated with an increased frequency of leg deformations and weakness [3,5,42,43]. In contrast, standing was observed most commonly in birds in the HO treatment. Standing time can be considered a measure of good leg health [45], but our result is consistent with other research [37,46,47], which suggests that an increase in animal density results in decreased opportunities for undisturbed rest. However, the reason for increased standing in HO is more likely to be because at low density there was reduced use of the range as a result of fearfulness [48] or adverse weather conditions [49].

Among the comfort behaviours, preening was observed more frequently in the low than high stocking density when no outdoor access was provided, although stocking density had no effect if outdoor access was available. Birds usually prefer to preen themselves indoors [21,50] and it can be considered a nonessential behaviour that is performed when immediate needs have been satisfied [9]. However, it is rarely seen under conditions of hardship [9,51], which explains why it was seen less in the birds at the high stocking density. It also progressively decreases with age, due to reduced physical activity, both for weight gain and reduction in space [52], as observed in the present study.

Outdoor access increased foraging, as expected, but only at the low indoor stocking density, demonstrating a reluctance of birds at the high density to forage. Foraging, as the appetitive phase of feeding [53], comprises a significant component of the time budget of jungle fowl, but in domestic chickens it tends to be reduced in unsuitable environments, such as when litter material or enrichment is absent [9]. Foraging is also a good indicator of a comfortable and less anxious state in chickens [54], suggesting that the high density did not allow for normal behaviour. Encouraging range use by providing diverse stimuli [50] may elicit foraging through activities like pecking, scratching, tearing, biting and harvesting of seeds [24,55,56]. This has implications for the design of outdoor access poultry systems, since tree cover as well as enrichment environment should be provided to stimulate ranging, increasing access to a diversity of plant species [57]. The use of wood shavings as litter, which was maintained by adding new wood shavings every week in this study, is likely to have been a source of enrichment [58].

### 4.2. Effects on Stress Indicators

The H/L ratio tends to be reduced in subtropical birds in the winter season [59], due to a decrease in temperature. It could be indicative of heat stress, particularly because the ratio was reduced in birds with outdoor access at low density on week 6. This may represent those most likely to experience heat stress, since the smaller birds in week 3, when ambient temperatures were hotter, as seen in Table 2, would have been less prone to heat stress. Chickens have greater floor space at a young age, which allows them to maintain homeostasis under high temperature (by conduction, convection and evaporative heat loss) [60]. Commercial broilers, such as the Hubbard Flex birds used in this study, are less tolerant to heat stress than tropical native breeds [61]. Temperatures above 30 °C and relative humidity levels above 80% inside the housing are likely to cause fast-growing chickens to experience heat stress [26]. Although our ambient conditions did not quite reach this level, it can be speculated that, in week 6, birds were particularly prone to heat stress. Heat stress can intensify the effects of overcrowding [62,63], hence it is not surprising that indoor only birds at the high stocking density in week 6 had increased H/L ratio, as a result of heterophilia.

Serum corticosterone levels were increased in week 6 of age when outdoor access was provided, and birds were kept at low density. Concentrations of corticosterone may increase in response to induced physical activity [64] or biological and production stressors like heat, cold, stocking density, restraint, cooping, and shackling [65]. In this context, fear is considered an important component of stress, which can lead to high serum corticosterone levels [66,67]. A lower percentage of chickens outside has been associated with fear [68]. Therefore, increased serum corticosterone levels could indicate that the time spent outdoors probably was not enough for the chickens at low stocking density to become habituated and, consequently, they were afraid. Elevation in serum corticosterone concentrations has been associated with an increase in H/L ratios due to leukopenia (lymphopenia) and heterophilia [69]. However, in the present study, both indicators were greater in week 6 of age but differed in terms of housing system, suggesting a direct relationship did not exist between H/L ratio and corticosterone. In previous research in poultry, H/L ratio has not been associated with an increase in glucocorticoid hormones in response to certain stressors [65,67].

Spleen and bursa weights and spleen relative weight all tended to be greater in chickens raised indoors than those raised with access to an outdoor area. Decreased weight of lymphoid organs in poultry has been associated with high levels of stress-induced serum corticosterone. Chronically elevated corticosterone concentrations elicit involution of lymphoid organs such as spleen and bursa of Fabricius by the depletion effect on lymphocytes from germinal cells with dysregulated immune responses, related to immunosuppression [15,17,70]. As discussed above, insufficient opportunities to habituate due to fear could have led birds to interpret stimuli outdoors as an environmental stressor, which caused an elevation in corticosterone concentrations and then, a decrease in bursa and spleen weight. However, bursa relative weight was greater in LI treatment, which can be explained by higher body weight in week 6 in this treatment.

### 4.3. Effects on Performance Indicators

Chickens raised indoors and at low density had higher body weight. In contrast, birds with outdoor access were more active with slower growth as a result. This effect could be explained by the greater frequency of feeding at low density and less locomotion when no outdoor access was provided, which was also reduced with increasing density. The density results in weeks 4 and 5 are in accordance with a previous study, which reported that body weight of broilers reared in cages with different stocking densities was increased as stocking density decreased [43]. Clearly there may be different responses between birds in battery cages (in this case 50 × 60 × 36 cm), which have limited opportunity to move, and uncaged birds, that have much more opportunity to move. The fact that both scenarios present similar results, suggest that the weight difference is not just due to increased activity in the uncaged birds, with birds in lower stocking density conceivably experiencing less stress in both systems.

## 5. Conclusions

This study assessed the effects of housing system and stocking density on behaviour and stress indicators of commercial-line broilers in a tropical region. The main effect of outdoor access on behaviour was to increase locomotion, which was greatly restricted when birds were kept at high stocking densities indoors, causing them to stand less, preen less and lie down more. Birds given outdoor access foraged more, but only if kept at low indoor stocking densities. Birds offered outdoor access showed evidence of reduced stress, although those kept at low stocking density inside and given outdoor access showed some evidence of fear at the end of the experiment. Birds with outdoor access generally gained less weight than those kept permanently indoors, reflecting increased activity.

## Figures and Tables

**Figure 1 animals-09-01016-f001:**
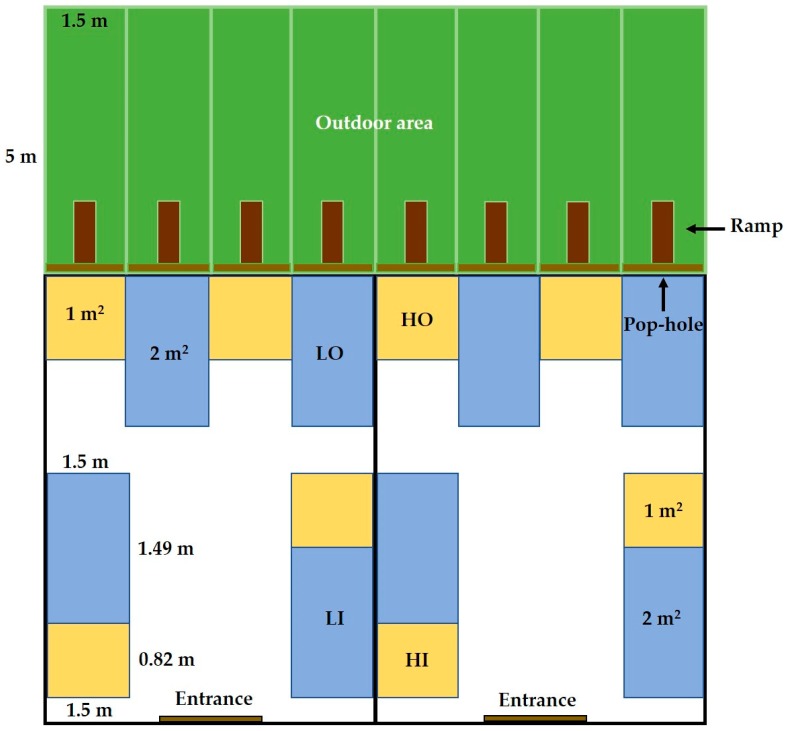
Experimental poultry facilities. Dimensions of pens include the supplementary space added to compensate for the area occupied by feeder and drinker. LO = Low stocking density with outdoor access; HO = High stocking density with outdoor access; LI = Low stocking density indoors; HI = High stocking density indoors.

**Figure 2 animals-09-01016-f002:**
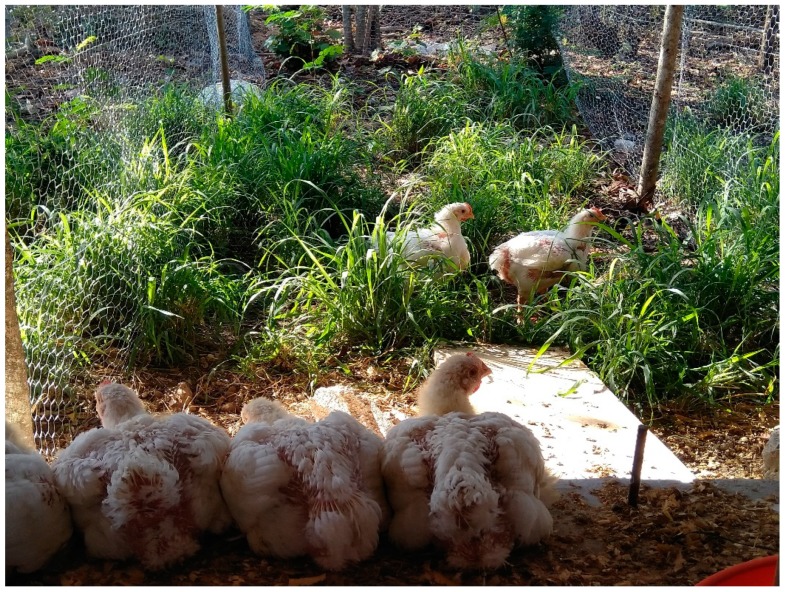
View of the outdoor area from inside a pen.

**Figure 3 animals-09-01016-f003:**
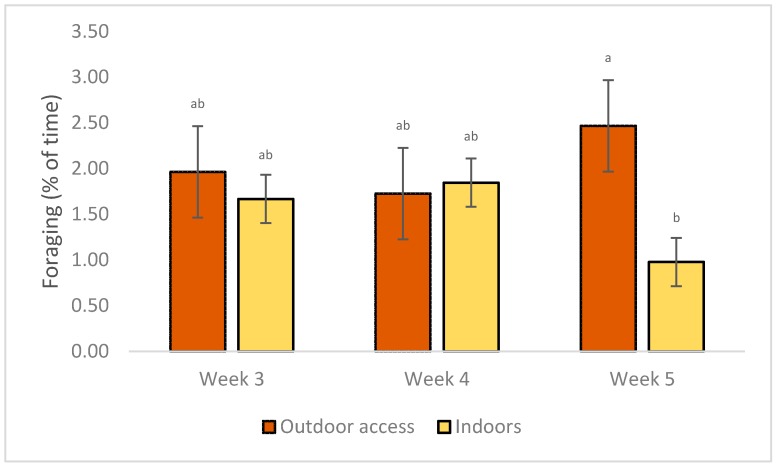
LS Means of the effects of system × week on foraging of commercial-line broilers. Different superscript letters indicate statistically significant differences (Tukey’s HSD test; *p* ≤ 0.05).

**Figure 4 animals-09-01016-f004:**
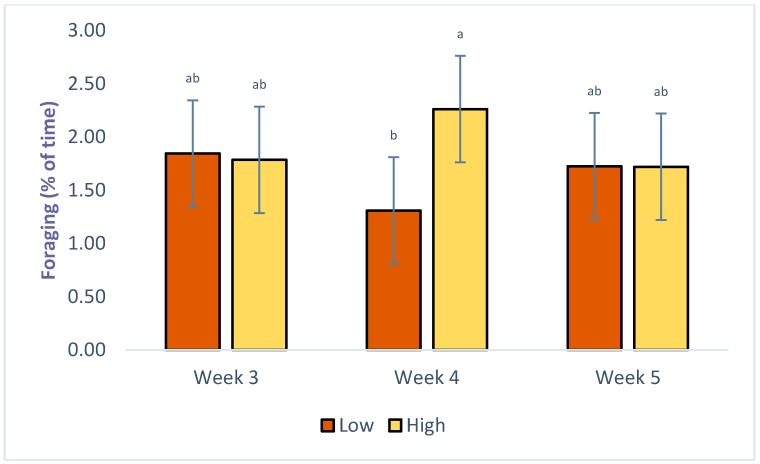
LS Means of the effects of density × week on foraging of commercial-line broilers. Different superscript letters indicate statistically significant differences (Tukey’s HSD test; *p* ≤ 0.05).

**Figure 5 animals-09-01016-f005:**
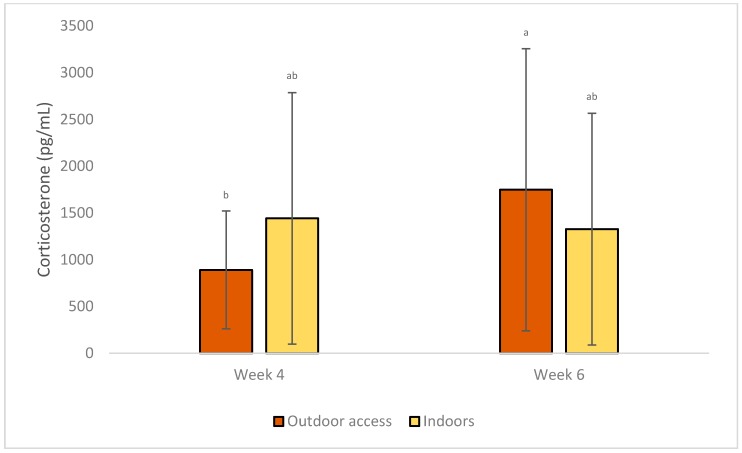
LS Means of the effects of system × week on corticosterone concentration of commercial-line broilers. Different superscript letters indicate statistically significant differences (Tukey’s HSD test; *p* ≤ 0.05).

**Figure 6 animals-09-01016-f006:**
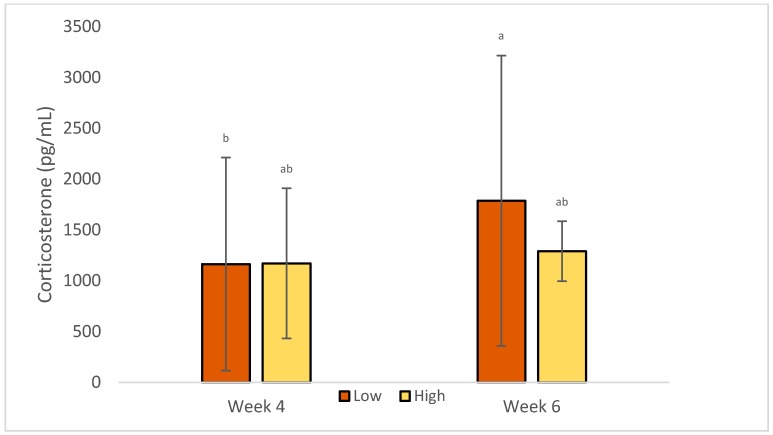
LS Means of the effects of density × week on corticosterone concentration of commercial-line broilers. Different superscript letters indicate statistically significant differences (Tukey’s HSD test; *p* ≤ 0.05).

**Figure 7 animals-09-01016-f007:**
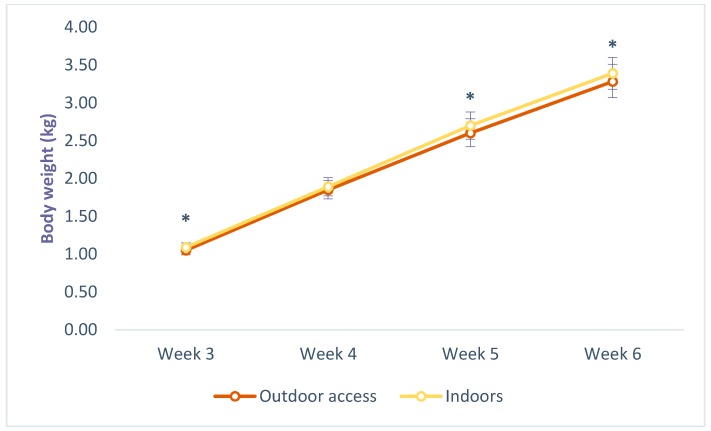
Effects of outdoor access on body weight of commercial-line broilers. * Indicates statistically significant differences (Tukey’s HSD test; *p* ≤ 0.05).

**Figure 8 animals-09-01016-f008:**
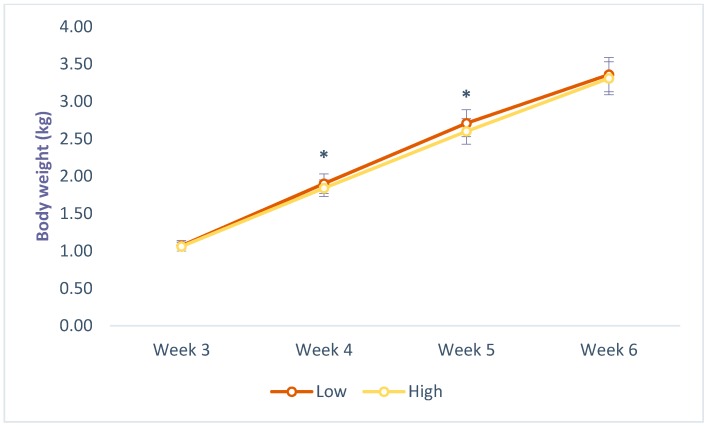
Effects of stocking density on body weight of commercial-line broilers. * Indicates statistically significant differences (Tukey’s HSD test; *p* ≤ 0.05).

**Table 1 animals-09-01016-t001:** Ethogram of 11 mutually exclusive behaviours, designed following a programme of ad libitum sampling of commercial-line broilers.

Category	Behaviour	Description
Individual	Feeding	Eating from food hopper, whilst standing, sitting or resting.
	Drinking	Drinking from the water trough, whilst standing, sitting or resting.
	Locomotion	Moving by walking or running.
	Lying	Main part of the body touching the ground, either chest or side.
	Standing	The abdomen not touching the litter or ground and the bird is motionless with no apparent movement of legs.
	Preening	Moving beak along the plumage
Interaction with the environment	Dustbathing	While lying with fluffed feathers, bird simultaneously and rapidly lifts its wings up and down multiple times, while scooping loose substrate material up into the feathers.
	Foraging	Scratching at the ground, both inside the pen and outdoors, with intermittent bouts of ground pecking at items (visible or not), usually followed by one or two steps backwards after a bout of ground scratching.
Social interaction	Mock fighting	After running toward each other, birds stop and stare at each other. The interaction is brief, harmless to the birds, and not directed persistently at any one bird.
	Grooming	The bird uses its beak to arrange the feathers of another bird.

**Table 2 animals-09-01016-t002:** Dry bulb temperature (°C) and relative humidity (RH, %) with standard deviations recorded during observations.

Hour	Outdoors				Indoors			
	Min (°C)	Max (°C)	Average (°C)	RH (%)	Min (°C)	Max (°C)	Average (°C)	RH (%)
Week 3								
7:00–8:00	19	25	21.6 ± 2.23	91 ± 6.3	22	27	24.3 ± 2.14	84 ± 6.4
12:00–13:00	32	35	34.0 ± 1.22	46 ± 9.7	32	35	33.8 ± 1.10	45 ± 8.7
16:00–17:00	29	32	30.7 ± 0.95	51 ± 3.2	30	33	31.3 ± 0.95	51 ± 3.0
Week 4								
7:00–8:00	18	25	23.0 ± 2.53	92 ± 4.8	21	26	25.0 ± 2.01	85 ± 1.9
12:00–13:00	27	28	27.7 ± 0.49	63 ± 1.4	28	28	28.0 ± 0.01	63 ± 1.4
16:00–17:00	24	27	25.6 ± 1.13	64 ± 2.3	26	28	27.1 ± 0.90	66 ± 4.0
Week 5								
7:00–8:00	17	24	20.6 ± 2.94	89 ± 8.2	20	24	21.9 ± 1.68	81 ± 6.8
12:00–13:00	29	30	29.7 ± 0.49	45 ± 1.1	30	30	30.0 ± 0.01	48 ± 1.0
16:00–17:00	23	28	26.0 ± 1.63	50 ± 3.2	26	26	26.9 ± 1.07	51 ± 4.8

**Table 3 animals-09-01016-t003:** Least Square Means of the effects of system, density and week of age on behaviour of commercial-line broilers ^1^.

Item	Feeding	Drinking	Locomotion *	Lying	Standing	Preening	Dustbathing *	Foraging
System								
Outdoor access	15.5 ± 3.26	6.24 ± 2.55	1.18 ± 0.78 ^a^	64.5 ± 4.12	3.48 ± 1.86	5.42 ± 1.90	0.85 ± 1.04	2.05 ± 0.68 ^a^
Indoors	15.4 ± 5.42	5.84 ± 1.50	0.79 ± 0.85 ^b^	66.4 ± 6.30	2.75 ± 1.10	5.80 ± 2.66	0.97 ± 0.77	1.50 ± 0.92 ^b^
Density								
Low	17.2 ± 4.84 ^a^	6.29 ± 1.81	1.29 ± 0.91 ^a^	62.8 ± 4.41 ^b^	2.62 ± 1.03	6.27 ± 2.25 ^a^	1.03 ± 0.86	1.63 ± 0.83
High	13.7 ± 3.20 ^b^	5.79 ± 2.34	0.68 ± 0.62 ^b^	68.1 ± 4.92 ^a^	3.61 ± 1.82	4.94 ± 2.19 ^b^	0.80 ± 0.96	1.92 ± 0.86
Week of age								
3	13.2 ± 3.09 ^b^	6.01 ± 1.68 ^a,b^	1.31 ± 0.78 ^a^	65.7 ± 4.81	2.95 ± 1.53	6.55 ± 1.92 ^a^	1.40 ± 1.12	1.81 ± 0.56
4	17.4 ± 5.47 ^a^	4.91 ± 2.07 ^b^	0.48 ± 0.74 ^b^	65.8 ± 5.65	2.83 ± 1.26	5.60 ± 2.26 ^a,b^	0.60 ± 0.66	1.79 ± 1.01
5	15.7 ± 3.56 ^a,b^	7.20 ± 1.92 ^a^	1.17 ± 0.76 ^a^	64.7 ± 5.81	3.57 ± 1.80	4.67 ± 2.43 ^b^	0.75 ± 0.71	1.72 ± 0.96
Main effect, *p*-value								
System	0.90	0.43	0.03	0.11	0.08	0.54	0.37	0.01
Density	0.004	0.33	0.01	<0.001	0.06	0.03	0.25	0.16
Week of age	0.02	0.003	<0.001	0.67	0.30	0.05	0.06	0.93
Interaction, *p*-value								
Sys × Den	0.06	0.39	0.01	<0.001	0.01	0.04	0.72	0.02
Sys × Week	0.70	0.12	0.60	0.15	0.93	0.12	0.57	0.008
Den × Week	0.40	0.31	0.60	0.01	0.50	0.79	0.55	0.006
Sys × Den × Week	0.65	0.04	0.88	0.44	0.89	0.69	0.95	0.21

^1^ Percentage of time performing each behaviour, based on a total of 210 min of observation. Sys = System; Den = Density; Week = Week of age. * Square root transformed for ANOVA. LS Means provided were calculated from original data, and those in the same column with different superscript letters indicate statistically significant differences (Tukey’s HSD test; *p* ≤ 0.05).

**Table 4 animals-09-01016-t004:** LS Means of the effects of system × density on behaviour of commercial-line broilers ^1^.

Behaviour	O		I	
	L	H	L	H
Feeding	16.4 *±* 3.00	14.7 *±* 3.41	18.1 *±* 6.20	12.7 *±* 2.78
Drinking	6.27 *±* 2.30	6.22 *±* 2.89	6.31 *±* 1.24	5.37 *±* 1.64
Locomotion	1.35 *±* 0.88 ^a,b^	1.01 *±* 0.87 ^a,b^	1.23 *±* 0.98 ^a^	0.36 *±* 0.36 ^b^
Lying	64.2 *±* 4.41 ^a,b^	64.8 *±* 3.98 ^a,b^	61.3 *±* 4.09 ^b^	71.4 *±* 3.30 ^a^
Standing	2.46 *±* 0.93 ^b^	4.49 *±* 2.04 ^a^	2.78 *±* 1.15 ^a,b^	2.72 *±* 1.03 ^a,b^
Preening	5.56 *±* 2.06 ^a,b^	5.28 *±* 1.81 ^a,b^	6.98 *±* 2.28 ^a^	4.60 *±* 2.56 ^b^
Dustbathing *	0.95 *±* 0.88	0.75 *±* 1.21	1.11 *±* 0.87	0.84 *±* 0.67
Foraging	2.06 *±* 0.80 ^a^	2.04 *±* 0.58 ^a^	1.19 *±* 0.63 ^b^	1.80 *±* 1.08 ^a,b^

^1^ Percentage of time performing each behaviour based on a total of 210 min of observation. O = Outdoor access; I = Indoors; L = Low stocking density; H = High stocking density. LS Means in the same row with different superscript letters indicate statistically significant differences (Tukey’s HSD test; *p* ≤ 0.05). * Square root transformed for ANOVA. LS Means provided were calculated from original data.

**Table 5 animals-09-01016-t005:** LS Means of the effects of system, density and week on stress indicators of commercial-line broilers.

Item	Corticosterone (pg/mL) ^1,^*	H/L Ratio ^2,^*	Heterophils (%) ^2,♦^	Lymphocytes (%) ^2,♦^
System				
Outdoor access	1319.9 ± 1200.6	1.75 ± 0.86 ^b^	52.6 ± 9.84	34.2 ± 9.48 ^a^
Indoors	1384.9 ± 1340.8	2.19 ± 1.08 ^a^	55.3 ± 14.5	28.3 ± 7.97 ^b^
Density				
Low	1474.8 ± 1460.8	2.06 ± 1.12	55.9 ± 12.14	32.6 ± 10.72
High	1230.0 ± 1017.1	1.89 ± 0.83	52.0 ± 12.21	29.9 ± 7.12
Week of age				
4	1166.8 ± 1140.3 ^b^	1.63 ± 0.80 ^b^	47.5 ± 10.8 ^b^	32.4 ± 8.54
6	1538.1 ± 1381.9 ^a^	2.31 ± 1.08 ^a^	60.5 ± 10.4 ^a^	30.1 ± 9.91
Main effect, *p*-value				
System	0.55	0.05	0.22	0.006
Density	0.95	0.83	0.07	0.21
Week of age	0.02	0.003	<0.001	0.27
Interaction, *p*-value				
Sys × Den	0.09	0.43	0.08	0.80
Sys × Week	0.01	0.003	<0.001	0.06
Den × Week	0.02	0.23	0.004	0.86
Sys × Den × Week	0.29	0.01	0.09	0.001

^1^ Based on a total of 32 duplicate samples in week 4 and 28 in week 6. ^2^ Based on a total of 32 samples in week 4 and 28 in week 6. **^♦^** Over the total white blood cell count. Sys = System; Den = Density; Week = Week of age. LS Means in the same column with different superscript letters indicate statistically significant differences (Tukey’s HSD test; *p* ≤ 0.05). * Log_10_ transformed for ANOVA. LS Means provided were calculated from original data.

**Table 6 animals-09-01016-t006:** LS Means of the effects of system × density × week on stress indicators of commercial-line broilers ^1^.

System	Density	Week	H/L Ratio	Heterophils (%) *	Lymphocytes (%) *
Outdoor	Low	4	2.05 ± 1.00 ^a,b^	56.6 ± 9.77 ^a,b^	31.2 ± 8.91 ^a,b^
		6	1.44 ± 0.95 ^b^	48.9 ± 9.31 ^b,c^	40.4 ± 11.4 ^a^
	High	4	1.46 ± 0.74 ^b^	46.4 ± 9.13 ^b,c^	35.6 ± 8.70 ^a,b^
		6	2.07 ± 0.61 ^a,b^	58.7 ± 6.26 ^a,b^	29.7 ± 5.40 ^a,b^
Indoors	Low	4	1.54 ± 0.90 ^b^	48.6 ± 11.2 ^b,c^	35.9 ± 9.33 ^a,b^
		6	3.21 ± 0.95 ^a^	69.7 ± 6.15 ^a^	22.8 ± 4.83 ^b^
	High	4	1.48 ± 0.45 ^b^	38.2 ± 4.59 ^c^	27.0 ± 4.54 ^b^
		6	2.54 ± 1.04 ^a,b^	64.6 ± 5.50 ^a^	27.6 ± 6.58 ^a,b^

^1^ Based on a total of 32 samples in week 4 and 28 in week 6. * Over the total white blood cell count. LS Means in the same column with different superscript letters indicate statistically significant differences (Tukey’s HSD test; *p* ≤ 0.05).

**Table 7 animals-09-01016-t007:** LS Means of the effects of system, density and week of age on lymphoid organs of commercial-line broilers ^1^.

Item	Spleen Weight (g)	Spleen RW (%)	Bursa Weight (g)	Bursa RW (%)
System				
Outdoor access	3.17 ± 0.64 ^b^	0.09 ± 0.02 ^b^	4.18 ± 1.05 ^b^	0.12 ± 0.03
Indoors	3.73 ± 0.85 ^a^	0.10 ± 0.02 ^a^	4.96 ± 1.06 ^a^	0.14 ± 0.03
Density				
Low	3.67 ± 0.89	0.10 ± 0.02	4.38 ± 1.02	0.12 ± 0.03
High	3.24 ± 0.67	0.09 ± 0.02	4.76 ± 1.20	0.14 ± 0.04
Main effect, *p*-value				
System	0.003	0.02	0.009	0.08
Density	0.08	0.17	0.32	0.15
Interaction, *p*-value				
Sys × Den	0.75	0.76	0.15	0.046

^1^ Based on a total of 32 samples. RW = Relative weight; Sys = System; Den = Density; Week = Week of age. LS Means in the same column with different superscript letters indicate statistically significant differences (Tukey’s HSD test; *p* ≤ 0.05).

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
