# Peer review of "Effects of Outdoor Access and Indoor Stocking Density on Behaviour and Stress in Broilers in the Subhumid Tropics"

_animals, 2019, doi:10.3390/ani9121016_

Round 1

Reviewer 1 Report

The authors raised broilers on litter in confinement with and without outdoor access at both low and high stocking density and evaluated stress response indicators and body weight gain.

One topic that has not been discussed is the sample size of 10 chickens related to a commercial house of 50 to 88,000 broilers/ house, especially in relationship to fearfulness.

Statements for a few references are not correctly paraphrased and need to be corrected.

Specific comments:

L37, L67 Reword since you did not measure “stress” only “stressor response”, indication reduced stressor response.

L38 Provided how you quantified “appeared more fearful”.

L41 Delete “free range” since the outdoor access provided “fenced pens” was not free range.

L48 Replace “Reducing” since to be reduced space, space would have to have been larger at a previous time. Suggest “less”, “smaller” or “crowded”.

L49 Delete “any” since a marginally smaller pen would not impact welfare.

L50-52 Consider that the “evolutionary microenvironment” included predators and a sporadic feed supply impact on welfare.

L52-54 Reference [7] Provide appropriate reference or delete. These authors used methods of reducing anti-nutritional factors in forages: Soaking, Heat Treatment, or Fermentation and fed the “diets for chicken layers” not outdoor access forages for broilers.

L65 Replace “from” with “some”.

L87 Justify a sample size of 10 broilers/pen with behavioral references “normal”.

L91, L126 Justify the use of “bell-drinkers” since they are no longer commonly used in commercial chicken production.

L92 Provide the lighting program that accompanied naturally-ventilated pens.

L197 Report RH in whole numbers by rounding off decimal points, since the most accurate instruments available read in whole numbers.

L206 Provide the a and b superscripts for foraging in Table 3 to indicate significant difference, p = 0.01.

L212 Correct statement since foraging did not differ for broilers provide outdoor access house at low (2.06a) of high (2.04a) density in Table 4.

L238, L240 Replace “reduced” with “lower” since you do not have before and after data to claim a reduction.

L273 vs. L163, L163 states “The spleen and bursa of Fabricius of eight animals randomly selected per treatment” but L273 states “Based on a total of 64 samples.” Eight animals from the 4 treatments equals a total of 32 samples not 64?

L269, L270, L293 Replace “increased” with “higher” since you only measured at week 6.

Replace “without outdoor access” with “housed indoors” as in the previous sentence.

L280 Complete statement by adding “compared to those housed at high density”.

L289-290 Provide an explanation for this sentence since the experiments were not in “a commercial environment” or delete.

L293-296 Reference Zhao [36] in Table 3 reported feeding frequency for indoor of 12.65% and for outdoor of 10.78% with a p = 0.021, which is significantly different at P < 0.05.

L316 Reference [46] should be replaced with the reference Edgar cited [Campler et al., 2009] since Edgar et al. did not evaluate fearfulness.

Campler M., Jongren M., Jensen P. 2009. Fearfulness in red junglefowl and domesticated white leghorn chickens. Behav. Process. 81, 39–4310.1016/j.beproc.2008.12.018 (doi:10.1016/j.beproc.2008.12.018)

L317 Correct statement since reference [47] does not support your implication that “fearfulness” is the reason that broilers do not venture outside the shelter during adverse weather.

“This result, together with the finding that the number of outside birds further than 5 m from the house was negatively associated with TI duration in trial 1, may indicate that fearfulness could affect the distance chickens travel from the house, and that chickens who spend more time outside also become less fearful over time. However, no relationship was found between percentage of animals outside and fearfulness, and when a relationship was found (birds further than 5 m from stable; differences between treatment groups), either the duration or the number of inductions was found to be related, but never both. To get more insight into this relationship, data on individual animals would be needed.”

L324 Correct statement since reference [46] does not support the statement that density influences preening “Similarly, for dustbathing and preening there was no clear effect between the different densities”.

L356-357 Correct statement since reference [47] does not support your implication that “fearfulness” is the reason that broilers do not venture outside the shelter.

L378 Complete the statement, which was also reduced with increasing “age”?

L379 Complete statement since Figure 8 presents no significant difference in body weight at 6 weeks between high or low stocking densities.

L380 Further discuss that the female broilers in reference [44] were housed in battery cages and the relationship to litter reared broilers. “Then, chicks were allotted to 12 cages in a 4 deck cage system to construct 4 replicates/treatment. Every cage had breadth of 3000 cm2 (50 cm length, 60 cm width and 36 cm depth).”

Author Response

Response letter to the reviewers' comments

Manuscript ID: animals-633012

Effects of outdoor access and indoor stocking density on behaviour and stress in broilers in the sub-humid tropics

November 07, 2019

We would like to thank the reviewers for their high quality and constructive reviews of our manuscript.

A detailed item-by-item response to each of the reviewers’ points follows. Our replies are marked in italic.

Reviewer 1

Comments and Suggestions for Authors

The authors raised broilers on litter in confinement with and without outdoor access at both low and high stocking density and evaluated stress response indicators and body weight gain.

One topic that has not been discussed is the sample size of 10 chickens related to a commercial house of 50 to 88,000 broilers/ house, especially in relationship to fearfulness.

Statements for a few references are not correctly paraphrased and need to be corrected.

Specific comments:

L37, L67 Reword since you did not measure “stress” only “stressor response”, indication reduced stressor response.

The sentence in line 37 was modified: “Outdoor access reduced heterophil/lymphocyte ratio, indicating reduced stressor response”.

L38 Provided how you quantified “appeared more fearful”.

Fearfulness was not measured in this study. The sentence in line 38 was modified: “Birds with low stocking density indoors and outdoor access appeared more responsive to stressors, with elevated corticosterone and reduced spleen and bursa weights (p < 0.05)”.

L41 Delete “free range” since the outdoor access provided “fenced pens” was not free range.

“Free range” was removed from key words (line 42).

L48 Replace “Reducing” since to be reduced space, space would have to have been larger at a previous time. Suggest “less”, “smaller” or “crowded”.

The word “Reducing” was changed with “crowded” (now in line 49): “Crowded living space can have a significant positive impact on farmer incomes, as these generally increase with density [2]”.

L49 Delete “any” since a marginally smaller pen would not impact welfare.

Sentence in line 49 (now in line 50) was changed to: “However, decrease in space allowance is likely to have detrimental effects on the welfare of the birds [3–5]”.

L50-52 Consider that the “evolutionary microenvironment” included predators and a sporadic feed supply impact on welfare.

Information in lines 50-52 (now in lines 51-57) was added to emphasise that outdoor access can have some disadvantages: “Given that the microclimate for chickens in most tropical conditions accords with their evolutionary environment, outdoor access could potentially improve their welfare without greatly increasing costs but it is still necessary to develop standards that ensure its proper functioning. [6]. In particular the possibility of predation, sporadic feed availability and transmission of diseases from wildlife must be recognised, but on the positive side, it could”.

L52-54 Reference [7] Provide appropriate reference or delete. These authors used methods of reducing anti-nutritional factors in forages: Soaking, Heat Treatment, or Fermentation and fed the “diets for chicken layers” not outdoor access forages for broilers.

The reference was removed.

L65 Replace “from” with “some”.

Now in line 68, this fragment was changed to: “avian immune response takes some time”.

L87 Justify a sample size of 10 broilers/pen with behavioral references “normal”.

Information justifying a group size of 10 birds per pen is now mentioned from line 90: “A group size of 10 birds per pen was maintained in all treatments, and two different enclosure sizes were used to achieve the two stocking densities. This constant group size has been showed to have only small effects on behaviour in terms of observed nearest-neighbour distances and amount of space used, compared with larger group sizes [27].

L91, L126 Justify the use of “bell-drinkers” since they are no longer commonly used in commercial chicken production.

Information regarding the use of bell-drinkers was added in line 104: “Bell-type drinkers were chosen over others since they provide a higher water availability and promote natural drinking behaviour [28]”.

L92 Provide the lighting program that accompanied naturally-ventilated pens.

Lighting program is now provided in line 106: “Lighting program was 24L:0D for the first seven days” and in lines 109-110: “Lighting program from day 15 to 21 was 14L:10D and 16L:8D from day 22 to the end of the experiment”.

L197 Report RH in whole numbers by rounding off decimal points, since the most accurate instruments available read in whole numbers.

Relative humidity data was converted to whole numbers, with one extra significant figure for the Standard Deviations, as is conventional.

L206 Provide the a and b superscripts for foraging in Table 3 to indicate significant difference, p = 0.01.

Superscripts for foraging in Table 3 were added.

L212 Correct statement since foraging did not differ for broilers provide outdoor access house at low (2.06a) of high (2.04a) density in Table 4.

“And decreased foraging” was removed from the statement (now in lines 230-231): “If there was an outdoor area, the high indoor density increased standing, compared with the low density”.

L238, L240 Replace “reduced” with “lower” since you do not have before and after data to claim a reduction.

The sentence was removed since it reports main effects instead of interactions, which were significant. This was a requirement of Reviewer 2.

L273 vs. L163, L163 states “The spleen and bursa of Fabricius of eight animals randomly selected per treatment” but L273 states “Based on a total of 64 samples.” Eight animals from the 4 treatments equals a total of 32 samples not 64?

Information regarding the number of samples was corrected (line 290): “Based on a total of 32 samples”.

L269, L270, L293 Replace “increased” with “higher” since you only measured at week 6.

“Increased” was replaced with “higher” (line 286: “Bursa weight was higher in birds housed indoors, with relative weight being particularly higher in the LI treatment (0.14 ± 0.03), compared to LO treatment (0.11 ± 0.04) (p = 0.046). Statement in line 293 was removed.

Replace “without outdoor access” with “housed indoors” as in the previous sentence.

“Without outdoor access” was replaced with “housed indoors” (line 286): “Bursa weight was higher in birds housed indoors, with relative weight being particularly higher in the LI treatment (0.14 ± 0.03), compared to LO treatment (0.11 ± 0.04) (p = 0.046).

L280 Complete statement by adding “compared to those housed at high density”.

“Compared to those housed at high density” was added to the statement (now in line 297): “and those kept at low density were heavier on weeks 4 (1.90 ± 0.13 vs. 1.84 ± 0.11; p = 0.004) and 5 (2.71 ± 0.18 vs. 2.60 ± 0.17; p = < 0.001), compared to those housed at high density (Figures 7 and 8)”.

L289-290 Provide an explanation for this sentence since the experiments were not in “a commercial environment” or delete.

The sentence in lines 289 and 290 was deleted.

L293-296 Reference Zhao [36] in Table 3 reported feeding frequency for indoor of 12.65% and for outdoor of 10.78% with a p = 0.021, which is significantly different at P < 0.05.

The whole paragraph was deleted since it discusses the main effects instead of interactions.

L316 Reference [46] should be replaced with the reference Edgar cited [Campler et al., 2009] since Edgar et al. did not evaluate fearfulness.

Campler M., Jongren M., Jensen P. 2009. Fearfulness in red junglefowl and domesticated white leghorn chickens. Behav. Process. 81, 39–4310.1016/j.beproc.2008.12.018 (doi:10.1016/j.beproc.2008.12.018)

Reference “Edgar et al. 2011” was replaced with “Campler et al., 2009” (line 332).

L317 Correct statement since reference [47] does not support your implication that “fearfulness” is the reason that broilers do not venture outside the shelter during adverse weather.

“This result, together with the finding that the number of outside birds further than 5 m from the house was negatively associated with TI duration in trial 1, may indicate that fearfulness could affect the distance chickens travel from the house, and that chickens who spend more time outside also become less fearful over time. However, no relationship was found between percentage of animals outside and fearfulness, and when a relationship was found (birds further than 5 m from stable; differences between treatment groups), either the duration or the number of inductions was found to be related, but never both. To get more insight into this relationship, data on individual animals would be needed.”

The statement was corrected by adding “or” to make clear that “fearfulness” and “adverse weather conditions” are two different factors (line 332).

L324 Correct statement since reference [46] does not support the statement that density influences preening “Similarly, for dustbathing and preening there was no clear effect between the different densities”.

The reference was changed with the correct one (line 340):

 Fortomaris, P.; Arsenos, G.; Tserveni-Gousi, A.; Yannakopoulos, A. Performance and behaviour of broiler chickens as affected by the housing system. Arch. fur Geflugelkd. 2007, 71, 97–104.

L356-357 Correct statement since reference [47] does not support your implication that “fearfulness” is the reason that broilers do not venture outside the shelter.

The reference was changed with the correct one (line 372):

Campbell, D.; Hinch, G.; Downing, J.; Lee, C. Fear and coping styles of outdoor-preferring, moderate-outdoor and indoor-preferring free-range laying hens. Appl. Anim. Behav. Sci. 2016, 185, 73–77.

L378 Complete the statement, which was also reduced with increasing “age”?

The statement was completed with “density”: “This effect could be explained by the greater frequency of feeding at low density and less locomotion when no outdoor access was provided, which was also reduced with increasing density” (line 407).

L379 Complete statement since Figure 8 presents no significant difference in body weight at 6 weeks between high or low stocking densities.

The statement was completed by adding “in weeks 4 and 5” (lines 394-396): “The density results in weeks 4 and 5 are in accordance with a previous study, which reported that body weight of broilers reared in cages with different stocking densities, was increased as stocking density decreased [43]”.

L380 Further discuss that the female broilers in reference [44] were housed in battery cages and the relationship to litter reared broilers. “Then, chicks were allotted to 12 cages in a 4 deck cage system to construct 4 replicates/treatment. Every cage had breadth of 3000 cm2 (50 cm length, 60 cm width and 36 cm depth).”

More details about the experiment cited were added (lines 394-401): “The density results in weeks 4 and 5 are in accordance with a previous study, which reported that body weight of broilers reared in cages with different stocking densities was increased as stocking density decreased [43]. Clearly there may be different responses between birds in battery cages (in this case 50 x 60 x 36 cm), which have limited opportunity to move, and uncaged birds, that have much more opportunity to move. The fact that both scenarios present similar results, suggest that the weight difference is not just due to increased activity in the uncaged birds, with birds in lower stocking density conceivably experiencing less stress in both systems”.

Reviewer 2 Report

This study complements other work on outdoor access of fast growing broilers in other climatic zones published in this journal. The experimental design is straightforward and the statistical analyses appropriate. However, the presentation of the results needs to be improved. Interactions between the main effects are not given the appropriate attention.

It is important to acknowledge that density and pen size are confounded in this study. It is not clear if density or pen size or their combination are the causes of the observed effects. It is quite logical that locomotion was influenced by pen size and this needs to be discussed.

The simple summary includes sentences that are not logically linked. Line 14: It does not follow from the first part of the sentence that chickens have the ability to cope with the new environment. Reading the introduction, I think you should substitute ‘new’ with ‘outdoor’.

The sentence starting at line 19 is not logical or it is incomplete. There seems to be an interaction but only one condition is mentioned, having higher stocking density without access. Does this mean that higher stocking density with no access to outdoors did not have an effect?

Line 28: How many chickens? Please mention the hybrid.

Line 30: Was the scan sampling done indoors or outdoors or both?

Line 39: 'Although' is the wrong word. Increased activity is probably the reason for slower growth because the birds devote more calories to locomotion.

Line 40: Growth is only mentioned in the conclusion.

Line 66: Add ‘but see …’.

Line 98: Were weights taken on the individual or the pen? Were the chicks individually marked?

Lines 116ff: Did you check the intra-observer reliability from the videos? Was the observer blinded when the videos were scored?

Fig. 1. Where were the windows? Did you check the temperature in different parts of the house to exclude that outdoor pens on one side had other temperatures?

What was the light regime? Did indoor birds have natural light?

Table 3. I am not sure how the observations were scored for birds with outdoor access. You observed them inside the barn and outside. Did you average the observations? Week 4 seems to be special. E.g. locomotion was about half during this week. Do you have an idea what could be the cause?

Lines 205ff: Whenever you have a significant interaction you cannot discuss the main effects. It does not make sense. So drop the differences in all traits except drinking and dustbathing and report the differences in the interactions in Table 4 instead.

Lines 206f: No. There were no differences between outdoor and indoor but between the 2 densities in indoor pens.

Lines 207f: No differences for feeding. Drop the entire sentence.

Lines 211ff. No. If there was an outdoor area there was no effect of indoor density on foraging, only on standing.

Lines 222f. Only in week 5 birds foraged more when outdoor access was provided compared with no outdoor access. For birds with outdoor access foraging did not differ among the weeks.

Lines 224f. Only in week 4 birds foraged less under low density conditions compared with high density conditions.

Lines 237ff. When there is a 3way interaction for HL ratio you cannot interpret the main effects. So concentrate on the results in Table 6 when discussing HL ratios.

Line 268. Write relative spleen weight?

Line 376. Was there an interaction between outdoor access and density? If yes, you have to interpret the interaction and not the main effects.

The discussion has to be adapted to the results considering the interactions.

Lines 303f. Mention that in this study density was confounded with pen size. I know that you either keep the size of the group constant (as you chose) or the size of the pen. Now you cannot say whether the differences are due to the density or the pen size. It is possible that birds move more when the pen is larger, this is realistic, and it might not have anything to do with density. You need to mention this.

Lines 325f. Where did the birds forage, indoors or outdoors or both?

Line 334. Can you say something about the quality of litter? Was it loose or was it too firm for scratching?

Lines 344ff. Provide a table with high temperatures during the weeks of the study. Were temperatures different in week 4?

Lines 355ff. Corticosterone also increases due to activity. Maybe outdoor birds were more active?

Line 365. weights instead of weight.

Author Response

Response letter to the reviewers' comments

Manuscript ID: animals-633012

Effects of outdoor access and indoor stocking density on behaviour and stress in broilers in the sub-humid tropics

November 07, 2019

We would like to thank the reviewers for their high quality and constructive reviews of our manuscript.

A detailed item-by-item response to each of the reviewers’ points follows. Our replies are marked in italic.

Reviewer 2

Comments and Suggestions for Authors

This study complements other work on outdoor access of fast growing broilers in other climatic zones published in this journal. The experimental design is straightforward and the statistical analyses appropriate. However, the presentation of the results needs to be improved. Interactions between the main effects are not given the appropriate attention.

It is important to acknowledge that density and pen size are confounded in this study. It is not clear if density or pen size or their combination are the causes of the observed effects. It is quite logical that locomotion was influenced by pen size and this needs to be discussed.

We have acknowledged this in the Materials and Methods (lines90-100) and Discussion (lines 321-323).

The simple summary includes sentences that are not logically linked. Line 14: It does not follow from the first part of the sentence that chickens have the ability to cope with the new environment. Reading the introduction, I think you should substitute ‘new’ with ‘outdoor’.

The word “outdoor” was added to make clear that stressors could be different in a human-controlled environment (line 14): “thus the ability to cope with outdoor environment may have been modified”.

The sentence starting at line 19 is not logical or it is incomplete. There seems to be an interaction but only one condition is mentioned, having higher stocking density without access. Does this mean that higher stocking density with no access to outdoors did not have an effect?

The sentence starting in line 19 was corrected by changing the behaviours affected. “With no outdoor access” was replaced with “with outdoor access”: “Some blood parameters indicated that outdoor access reduced stress, but the behavioural effects of providing outdoor access depended on indoor stocking density, with most standing in birds at the high stocking density with outdoor access”.

Line 28: How many chickens? Please mention the hybrid.

Information regarding the number of chickens and the hybrid was added (line 28): “One hundred and sixty chickens were assigned to one of four treatments in a factorial design: with or without outdoor access and low or high stocking density indoors”. And at line 111 Hubbard Flex was added.

Line 30: Was the scan sampling done indoors or outdoors or both?

“Both indoors and outdoors” was added to complete the statement (now in lines 30-32): “Scan sampling was used to record the number of birds engaged in each activity of this ethogram, both indoors and outdoors”.

Line 39: 'Although' is the wrong word. Increased activity is probably the reason for slower growth because the birds devote more calories to locomotion.

The word “Although” was deleted (line 39): “There were welfare benefits of outdoor access, principally in terms of increased activity”.

Line 40: Growth is only mentioned in the conclusion.

Aside from Simple summary, Abstract and Introduction, “growth” is now mentioned in the Discussion regarding performance indicators (lines 391-392): “Chickens raised indoors and at low density had higher body weight. In contrast, birds with outdoor access were more active with slower growth as a result”.

Line 66: Add ‘but see …’.

Authors were not clear on the appropriate place in the statement to add the words requested. We’ve added it on line 69, “but see 13-16”.

Line 98: Were weights taken on the individual or the pen? Were the chicks individually marked?

The weight of all chickens was recorded, but it was reported weekly as the average per treatment. Chickens were not identified for this purpose (lines 114-115): “The body weight of all chickens was recorded weekly as the average per treatment. Individual birds were not identified for this purpose”.

Lines 116ff: Did you check the intra-observer reliability from the videos? Was the observer blinded when the videos were scored?

Yes, it was checked. Information regarding intra-observer reliability was added in lines 152-153: “The sole observer was blind to all aspects of treatments and intra-observer reliability, expressed as Pearson correlation coefficient, was 0.99 [30]”.

Fig. 1. Where were the windows? Did you check the temperature in different parts of the house to exclude that outdoor pens on one side had other temperatures?

Pop-holes are marked in Figure 1. Detailed information about how indoor temperatures were measured was added (lines 156-157): “For outdoor access treatments, indoor temperatures were measured at different points of the pens, far from the pop-holes”.

What was the light regime? Did indoor birds have natural light?

Lighting program is now provided in line 106: “Lighting program was 24L:0D for the first seven days” and in lines 109-110: “Lighting program from day 15 to 21 was 14L:10D and 16L:8D from day 22 to the end of the experiment. The light regime was similar in indoors treatments”.

Table 3. I am not sure how the observations were scored for birds with outdoor access. You observed them inside the barn and outside. Did you average the observations? Week 4 seems to be special. E.g. locomotion was about half during this week. Do you have an idea what could be the cause?

The behaviour in outdoor access treatments was observed inside and outside the pens, without considering the location of birds. Information regarding this point was added in line 148: “For outdoor access treatments, behaviour was recorded without considering if birds were inside or outside the pen”.

In week 4 average midday temperatures were lower than weeks 3 and 5. However, dry bulb temperature was not sufficiently independent to be included in the analysis as a factor in this study, so it can’t be stated that it was the cause of these changes.

Lines 205ff: Whenever you have a significant interaction you cannot discuss the main effects. It does not make sense. So drop the differences in all traits except drinking and dustbathing and report the differences in the interactions in Table 4 instead.

Results of main effects were removed from the text and now are focused on interactions (lines 228-231): “Significant interactions between system and stocking density were observed (Table 4). Locomotion and preening were decreased in birds at the high stocking density in indoor pens. If there was an outdoor area, the high indoor density increased standing, compared with the low density”.

Lines 206f: No. There were no differences between outdoor and indoor but between the 2 densities in indoor pens.

The sentence that states the differences between outdoor and indoor for locomotion and foraging was removed.

Lines 207f: No differences for feeding. Drop the entire sentence.

The relevant text is “and there was a tendency for feeding to be also reduced in this treatment (p = 0.06)” (lines 229-230). It is accepted by most scientists that it is acceptable to refer to ‘trends’ when P > 0.05 and < 0.10. There is no biological sense in a strict cut-off point of P = 0.05, so we have respectfully left this in.

Lines 211ff. No. If there was an outdoor area there was no effect of indoor density on foraging, only on standing.

“Foraging” was deleted from the sentence (lines 230-231): “If there was an outdoor area, the high indoor density increased standing, compared with the low density”.

Lines 222f. Only in week 5 birds foraged more when outdoor access was provided compared with no outdoor access. For birds with outdoor access foraging did not differ among the weeks.

The sentence was modified to make it clearer (lines 240-241): “Birds foraged more frequently on week 5 than 3 or 4, if outdoor access was provided, compared with no outdoor access (Figure 3)”.

Lines 224f. Only in week 4 birds foraged less under low density conditions compared with high density conditions.

The sentence was modified to make it clearer (lines 242-244): “chickens in the high density treatments spent less time lying (66.04 ± 6.44 vs. 69.35 ± 3.52, p = 0.01) in week 5 than week 4, and foraged less (1.72 ± 0.94 vs. 2.26 ± 0.94, p = 0.006) in week 4 (Figure 4)”.

Lines 237ff. When there is a 3way interaction for HL ratio you cannot interpret the main effects. So concentrate on the results in Table 6 when discussing HL ratios.

Main effects results were removed to report only the information about the 3 way interaction (lines 257-262).

Line 268. Write relative spleen weight?

The word “spleen” was added before “relative weight”, now in line 285: “Spleen weight and spleen relative weight were greater in chickens raised indoors (Table 7)”.

Line 376. Was there an interaction between outdoor access and density? If yes, you have to interpret the interaction and not the main effects.

There was not an interaction between system and density for body weight. A sentence containing this information was added in line 298: “No significant interaction effects were found between system and density”.

The discussion has to be adapted to the results considering the interactions.

Lines 303f. Mention that in this study density was confounded with pen size. I know that you either keep the size of the group constant (as you chose) or the size of the pen. Now you cannot say whether the differences are due to the density or the pen size. It is possible that birds move more when the pen is larger, this is realistic, and it might not have anything to do with density. You need to mention this.

A paragraph acknowledging this situation was added in lines 92-100: “. In stocking density experiments either the group size must be kept constant and the total area varies, or group size differs, and the total area can be kept constant. We chose the former as group size variation at these low levels was deemed more likely to induce changes in behaviour than varied total area. However, it has been demonstrated that total area available can have more effects on locomotion and use of space than group size, but only with a tripling of total area, not the doubling in our study [27]. Hence it is acknowledged that stocking density effects may be confounded with total area for the group in this study.”; in lines 318-320: “Therefore, different pen sizes could also have exerted an effect on locomotion, since it has been found that greater enclosures sizes promote higher rates of movement and activity [40]” and in lines 321-323: “Additionally pen size is confounded with stocking density in this experiment, with larger pen sizes potentially increasing bird locomotion and inter-individual distance, as noted in the Method section [27,40]”.

Lines 325f. Where did the birds forage, indoors or outdoors or both?

Foraging was measured both indoors and outdoors. A sentence to make it clear was added in table 1.

Line 334. Can you say something about the quality of litter? Was it loose or was it too firm for scratching?

Information of how the litter was maintained dry was added in lines 350-352: “The use of wood shavings as litter, which was maintained by adding new wood shavings every week in this study, is likely to have been a source of enrichment [58]”.

Lines 344ff. Provide a table with high temperatures during the weeks of the study. Were temperatures different in week 4?

Table 2 was modified by adding minimum and maximum temperatures. In week 4, average midday temperatures were below average.

Lines 355ff. Corticosterone also increases due to activity. Maybe outdoor birds were more active?

A reference was added to mention that (induced) higher activity could be a cause of increased corticosterone concentration (lines 368-370): “Concentrations of corticosterone may increase in response to induced physical activity [64] or biological and production stressors like heat, cold, stocking density, restraint, cooping, and shackling [65]”.

Line 365. weights instead of weight

An “s” was added to “weight” (line 380): “Spleen and bursa weights and spleen relative weight all tended to be greater in chickens raised indoors than those raised with access to an outdoor area”.

Round 2

Reviewer 2 Report

The revised version is much improved and I have no further comments. All my concerns were satisfactorily addressed and resolved. This is a very interesting study on ranging in broilers.

Author Response

Manuscript ID: animals-633012

Effects of outdoor access and indoor stocking density on behaviour and stress in broilers in the sub-humid tropics

November 19, 2019

A detailed item-by-item response to each of the reviewers’ points follows. Our replies are marked in italic.

Line 21: “with most standing in birds”, does not make sense in this sentence. Please revise.

Revised to ‘birds at the high stocking density with outdoor access were found standing for longest and those at the high stocking density and no outdoor access were observed walking less and lying more than the other three groups.’.

Line 34: This sentence is not clear. Outdoor access caused less standing- or high stocking densities indoors- or both caused less standing, pressing and increased lying behavior?

Revised to ‘In an interaction between stocking density and outdoor access, birds at the high stocking density with no outdoor pens spent least time walking and preening and most time lying (p < 0.05)’ (line 35).

Line 50- What increases with density? This sentence is not still not clear.

Changed to: ‘Crowded living space can have a significant positive impact on farmer incomes, which generally increase with stocking density’.

Line 54-To ensure that what is properly functioning? 

Added the following (line 55): ‘the outdoor area is properly functioning as a form of enrichment for the birds’.

Line 64- Consider revising: ‘displaying certain behaviors’ and not “spent in various behaviors”.

Changed (line 66).

Line 90- This is still not clear. The 10 birds were kept in different enclosures than the 160? Or are you saying that not all the 160 birds had outdoor access- just 10 birds per treatment? Please consider revising this section again to make it clear.

Changed to: ‘Of the 40 birds in each treatment, these were allocated at random to four groups (replicates) of ten birds. Two different enclosure sizes…’ (line 92).

Line 109- Was there any natural lighting in the house? What was the light intensity in the house over the period of the study? Provision of appropriate lighting can affect productivity and health of broiler chickens (Olanrewaju et al., 2006). Low intensities have also been associated with reduced walking and standing (Buyse el al., 1996). Whereas high light intensities decrease body weight due to increase activity (Olanrewaju et al., 2006).

Yes, the two buildings in which the pens were constructed allowed the access to natural light. Lighting programs were completed with one 40 W incandescent bulb. Information was added in line 110: ‘The facilities were designed to allow the access to an average of 11 hours of natural light a day (6:00-17:00 hours). The remaining light hours were provided by using one 40 W incandescent bulb (15 lumens/watt) per building’ (line 112).

Line 131: Were there any trees/shade? The provision of shade structures will promote birds to go outside, as cover (tall shrubs or any other cover) provides protection from predators. It is the natural instinct of birds to find areas with such protection. If this is not provided, you do not have a true outdoor area that is suitable for birds to feel comfortable in. This may affect the number of birds that go outside and the duration.

Yes, there were. Information regarding cover in outdoor areas can be found in line 134: ‘The eight outdoor areas (5 x 1.50 m2 for each pen) were covered with natural vegetation (mostly Pennisetum ciliare and Leucaena leucocephala) (Figure 2)’.
